# Poor Physical Performance Is Associated with Postoperative Complications and Mortality in Preoperative Patients with Colorectal Cancer

**DOI:** 10.3390/nu14071484

**Published:** 2022-04-02

**Authors:** Francisco José Sánchez-Torralvo, Iván González-Poveda, María García-Olivares, Nuria Porras, Montserrat Gonzalo-Marín, María José Tapia, Santiago Mera-Velasco, José Antonio Toval-Mata, Manuel Ruiz-López, Joaquín Carrasco-Campos, Julio Santoyo-Santoyo, Gabriel Olveira

**Affiliations:** 1Unidad de Gestión Clínica de Endocrinología y Nutrición, Hospital Regional Universitario de Málaga, 29007 Malaga, Spain; fransancheztorralvo@gmail.com (F.J.S.-T.); mery.garcia.96@gmail.com (M.G.-O.); nporrasperez@hotmail.com (N.P.); montserratgonzalomarin@gmail.com (M.G.-M.); mjtapiague@gmail.com (M.J.T.); 2Instituto de Investigación Biomédica de Málaga (IBIMA), 29010 Malaga, Spain; 3Departamento de Medicina y Dermatología, Facultad de Medicina, University of Malaga, 29010 Malaga, Spain; 4Unidad de Gestión Clínica de Cirugía General y Digestiva, Hospital Regional Universitario de Málaga, 29010 Malaga, Spain; ivangpoveda@gmail.com (I.G.-P.); s.meravelasco@gmail.com (S.M.-V.); josetoval1@gmail.com (J.A.T.-M.); manuel.ruiz.lop@gmail.com (M.R.-L.); joaquincarrascocampos@gmail.com (J.C.-C.); julio.santoyo.sspa@juntadeandalucia.es (J.S.-S.); 5Centro de Investigación Biomédica en Red de Diabetes y Enfermedades Metabólicas Asociadas (CIBERDEM), Instituto de Salud Carlos III, 28029 Madrid, Spain

**Keywords:** physical performance, malnutrition, cancer, oncology, colorectal, surgery

## Abstract

Introduction: Poor physical performance has been shown to be a good predictor of complications in some pathologies. The objective of our study was to evaluate, in patients with colorectal neoplasia prior to surgery, physical performance and its relationship with postoperative complications and in-hospital mortality, at 1 month and at 6 months. Methods: We conducted a prospective study on patients with preoperative colorectal neoplasia, between October 2018 and July 2021. Physical performance was evaluated using the Short Physical Performance Battery (SPPB) test and hand grip strength (HGS). For a decrease in physical performance, SPPB < 10 points or HGS below the EWGSOP2 cut-off points was considered. Nutritional status was evaluated using subjective global assessment (SGA). The prevalence of postoperative complications and mortality during admission, at 1 month, and at 6 months was evaluated. Results: A total of 296 patients, mean age 60.4 ± 12.8 years, 59.3% male, were evaluated. The mean BMI was 27.6 ± 5.1 kg/m^2^. The mean total SPPB score was 10.57 ± 2.07 points. A total of 69 patients presented a low SPPB score (23.3%). Hand grip strength showed a mean value of 33.1 ± 8.5 kg/m^2^ for men and 20.7 ± 4.3 kg/m^2^ for women. A total of 58 patients presented low HGS (19.6%). SGA found 40.2% (119) of patients with normal nourishment, 32.4% (96) with moderate malnutrition, and 27.4% (81) with severe malnutrition. Postoperative complications were more frequent in patients with a low SPPB score (60.3% vs. 38.6%; *p* = 0.002) and low HGS (64.9% vs. 39.3%, *p* = 0.001). A low SPPB test score (OR 2.57, 95% CI 1.37–4.79, *p* = 0.003) and low HGS (OR 2.69, 95% CI 1.37–5.29, *p* = 0.004) were associated with a higher risk of postoperative complications after adjusting for tumor stage and age. Patients with a low SPPB score presented an increase in in-hospital mortality (8.7% vs. 0.9%; *p* = 0.021), at 1 month (8.7% vs. 1.3%; *p* = 0.002) and at 6 months (13.1% vs. 2.2%, *p* < 0.001). Patients with low HGS presented an increase in mortality at 6 months (10.5% vs. 3.3%; *p* = 0.022). Conclusions: The decrease in physical performance, evaluated by the SPPB test or hand grip strength, was elevated in patients with colorectal cancer prior to surgery and was related to an increase in postoperative complications and mortality.

## 1. Introduction

Colorectal cancer is the second most deadly and third most commonly diagnosed cancer in the world, and its global incidence and mortality are likely to increase in the coming decades. Surgery is the priority approach in most cases, especially in the early stages [1].

The oncological process and the surgery itself involve an inflammation process that leads to a metabolic stress response [2,3]. Patients with cancer present a degree of malnutrition due to their own underlying disease that conditions greater morbidity and mortality [4]. In addition, during surgery, a hypermetabolic response is produced with great catabolism, which leads to a nutritional and immune deficit [5]. In the response to this stress situation, the nutritional status and functional reserve are particularly relevant.

The prevalence of frailty in cancer patients undergoing colorectal surgery was 22.7% in a recent study [6]. Some studies have observed that a poor functional performance in the preoperative period is associated with higher postoperative morbidity [7], whereas an improvement in functional performance is related to a decrease in postoperative complications [8].

One of the most used tools for assessing functional capacity is the Short Performance Battery Test (SPPB). The SPPB is an effective tool for the assessment of physical function, developed by Guralnik [9]. This tool combines measurements of balance (standing, tandem, and semi-tandem), gait (4 m gait speed), strength, and endurance (rising from chair). Its score correlates significantly with institutionalization and mortality. It is validated to detect frailty and has high reliability in predicting disability [10].

On the other hand, about 35% of patients undergoing colorectal surgery are moderately to severely malnourished before surgery [11]. Malnutrition significantly affects the evolution of the surgical process with an unfavorable impact on the gastrointestinal tract, the endocrinological and immune systems, and cardiorespiratory function; delays wound healing, which implies an increase in morbidity, mortality, and hospital stay, with a consequent increase in health care costs [12].

As far as we know, the influence of preoperative physical performance using the SPPB and hand grip strength on postoperative complications and mortality in patients undergoing colorectal cancer has not been investigated.

Our hypothesis is that poor preoperative physical performance evaluated by the SPPB test and HGS in these patients is associated with postoperative complications and mortality regardless of nutritional status.

Hence, the aim of the present study was to evaluate the impact of preoperative physical performance evaluated by SPPB test and HGS on postoperative complications and mortality in patients undergoing colorectal cancer surgery.

## 2. Materials and Methods

We conducted a prospective study on patients with a diagnosis of colorectal cancer, between October 2018 and July 2021. All the patients proposed for intervention at the Coloproctology Unit of the Hospital Regional Universitario de Malaga were evaluated. All assessed patients who signed the informed consent were included. Data about the type of neoplasm and tumor stage were collected.

### 2.1. Assessment of Nutritional Status

A nutritional assessment which included subjective global assessment (SGA) [13] was performed. The following anthropometric measures were obtained: weight, height, and BMI. Height was calculated at baseline with a stadiometer (Holtain Limited, Crymych, UK), and weight was calculated with a weighing scale adjusted to 0.1 kg (SECA 665, Hamburg, Germany). Brachial circumference was measured using a flexible and non-elastic tape. Fasting blood was collected from the pre-anesthetic study, including albumin and C-reactive protein (CRP).

Malnourished or at-risk patients received a nutritional intervention appropriate to routine clinical practice [2].

### 2.2. Physical Performance

The Short Physical Performance Battery (SPPB) test was used to measure physical performance [9]. The SPPB test comprises three items: standing balance, walking speed, and chair stands. Each item was evaluated on a scale from 0 (inability to complete) to 4 (best performance possible).

In accordance with previous studies [14,15,16], patients were classified into low physical performance-related risk (SPPB total score < 10) or no physical performance-related risk (SPPB total score ≥ 10).

Hand grip strength was measured using the dominant hand with a Jamar dynamometer (Asimow Engineering Co., Los Angeles, CA, USA). For this test, the patients were sat comfortably with the shoulder adducted, the forearm neutrally rotated, the elbow flexed to 90°, and the forearm and wrist in a neutral position. They were told to perform 3 consecutive contractions one minute apart from each other, and the mean value was calculated.

### 2.3. Clinical Outcomes

Data concerning postoperative complications [17] and mortality during admission, at 1 month, and at 6 months were collected.

### 2.4. Data Analysis

Quantitative variables were expressed as the mean ± standard deviation. Comparison between qualitative variables was conducted via a chi-square test, with Fisher correction if necessary. The quantitative variable distribution was assessed by the Kolmogorov–Smirnov test. Differences between quantitative variables were analyzed using Student’s *t*-test and, for variables not following a normal distribution, using non-parametric tests (Mann–Whitney or Kruskal–Wallis). We designed multivariate logistic regression models in which the dependent variable was the presence of postoperative complications, also controlling for sex, age, and stage of tumor. For calculations, significance was set at *p* < 0.05 for two tails. The data analysis was performed with the SPSS 22.0 program (SPSS Inc., Chicago, IL, USA, 2013).

### 2.5. Ethics

The Provincial Research Ethics Committee of Málaga approved the study and informed consent was obtained from all participants. The ethical principles included in the latest revision of the Declaration of Helsinki and good clinical practice standards were applied.

## 3. Results

A total of 296 patients were evaluated (Figure 1). Their mean age was of 60.4 ± 12.8 years, and 59.3% of them were male. Their general features are displayed in Table 1.

Colon cancer was the most frequent type of cancer (60.8%), while most of the patients were at stages II and III (75.4%).

The mean BMI was 27.6 ± 5.1 kg/m^2^. More than half of the patients (62.5%) were overweight, and 24.3% of them were obese.

The results of the functional tests are shown in Table 2. No significant differences were found in the SPPB total score between genders.

After the categorization of patients according to their physical performance, 227 patients (76.7%) presented a normal score in the SPPB test (≥10), and 69 (23.3%) presented a low score (<10). The age of the low-SPPB group (73.1 ± 10 years) was significantly higher than that of the high-SPPB group (67.1 ± 9.9 years, *p* < 0.001), and a low SPPB score was more frequent in women (29.4% vs. 18.7%, *p* = 0.047). There were no significant differences in the type of cancer (colon or rectum, *p* < 0.257) or in the stage of tumor (*p* = 0.95). The high-SPPB group had a greater length of stay than the low-SPPB group (*p* = 0.018).

A total of 58 patients (19.6%) presented low hand grip strength, and 238 patients (80.4%) presented normal values.

Differences between the SPPB and hand grip strength groups are shown in Table 3.

SGA found 40.2% (119) of patients with normal nourishment, 32.4% (96) with moderate malnutrition, and 27.4% (81) with severe malnutrition (59.8% with malnutrition or at risk). Malnourished patients presented a low SPPB score (28.3% vs. 15.4%, *p* = 0.011) and low dynamometry (24.4% vs. 11.8%, *p* = 0.007) more frequently.

Postoperative complications were more frequent in patients with a low SPPB score (60.3% vs. 38.6%; *p* = 0.002) and low hand grip strength (64.9% vs. 39.3%, *p* = 0.001). No difference was found in the frequency of postoperative complications or mortality in malnourished patients according to SGA (*p* = 0.86).

Multivariable logistic regression analyses showed that a low SPPB score and low hand grip strength were associated with a higher risk of postoperative complications after adjusting for confounding variables such as age, gender, and stage of tumor (Table 4).

During hospital admission, eight patients (2.7%) died, increasing to nine (3%) one month after the intervention. After six months, 4.7% (14) of the patients were deceased.

An increased risk of mortality was found among patients with a low SPPB score and low hand grip strength (Table 5). No differences were found in mortality at any time according to malnutrition.

## 4. Discussion

To our knowledge, this is the first study to evaluate the influence of preoperative physical performance using the SPPB and hand grip strength on postoperative complications and mortality in patients undergoing colorectal cancer surgery. Our main finding was that a low preoperative physical performance was frequent and was associated with postoperative complications and mortality.

The Short Physical Performance Battery (SPPB) is a useful, three-part assessment and well-established tool for evaluating physical performance [9,18]. It evaluates three physical measurements that include standing balance, walking speed, and chair stands. In this way, it assesses different aspects of physical performance and function of the lower extremities, making it an excellent tool to identify frailty in adults [19]. The SBBP has many advantages: it requires little training to carry it out, takes only a few minutes to complete, and can be performed in a small space. Further, the results are reproducible and sensitive to changes in functionality through time [20]. Previous studies showed a significant trend toward age-related functional decline, with some differences between men and women [20]. In our study, patients with a low SPPB score were older than those with SPPB > 10, but no significant differences were found in the SPPB total score between genders. Despite the obvious relationship between the loss of functional capacity and age, the permanence of the effect of a low SPPB score on clinical outcomes after adjusting the logistic regression for age reinforces the importance of the functional status of these patients.

In our results, a low SPPB score was correlated with a higher frequency of postoperative complications. In a Japanese study, it was found that a poor physical performance measured by the SPPB test can be predictive of postoperative pulmonary complications after lung resection surgery [21]. Similarly, an impaired preoperative physical performance determined by the SPPB has been associated with a worse postoperative outcome after cardiac surgery, pancreaticoduodenectomy, and lung and kidney transplant surgeries [14,15,16,21]. A systematic review [22] showed that physical performance tests, including the SPPB, seem to correlate significantly with survival, which is consistent with the results of our study.

Hand grip strength is often cited as an indirect measure of malnutrition and a reliable prognostic tool [23] and correlates well with fat-free mass [24]. Low hand grip strength is associated with aging, but regardless of this relationship, it has been shown to be a powerful predictor of disability, morbidity, and mortality, and, by itself, a good marker of frailty [25]. Along those lines, in the present study, we found that low preoperative HGS is a good predictor of postoperative complications and mortality, with results comparable to those found with the SPPB test.

In our sample, 62.5% of the patients were overweight, and 24.3% of them were obese, exceeding series previously described in similar patients [26]. Only 8.1% of the subjects were below the established cut-off points for a low BMI [27]. The presence of a high percentage of patients with impaired physical performance and a high prevalence of overweight suggests a significant presence of sarcopenic obesity, an entity that is difficult to diagnose without morphofunctional tests in patients in the early stages of the tumor. The presence of both obesity and sarcopenia leads to particularly bad clinical outcomes. A meta-analysis showed that the presence of sarcopenia was associated with an increased risk of complications after gastrointestinal tumor resection [28], and sarcopenic obesity has been specifically associated with a lower survival rate in several populations [28,29].

The prevalence of malnutrition in our study was high at 59.8%; this figure is similar to others previously found [4]. SGA has been used previously in patients with colorectal cancer, finding a prevalence between 35 and 40% [11,30]. A Japanese study found a prevalence of preoperative malnutrition of 23.6%, which was associated with postoperative complications, overall survival, and disease-free survival in colorectal cancer patients after radical resection surgery [31]. These findings were also corroborated by the results from a large population database from the United States [32].

A worse nutritional status is associated with worse functional walking capacity and hand grip strength [26], but some studies have shown that functional assessment techniques such as hand grip strength are independent predictors of clinical outcomes regardless of the diagnosis of malnutrition, so these techniques provide an additional predictive value. These results suggest that the use of malnutrition and physical performance tools in combination may be valuable in hospital settings [33]. Although the close relationship between malnutrition and postoperative outcomes seems evident, in our study, no relationship was found between the two. Although causal relationships cannot be established due to the study design, we postulate that this relationship was interfered with by the nutritional intervention performed in all patients with malnutrition or at risk [34].

Malnutrition and physical performance screening is not routinely practiced before surgery [26]. Our findings support the use of the SPPB and HGS to evaluate physical function in cancer patients in addition to malnutrition screening before colorectal cancer surgery.

An early awareness of impaired physical performance and nutritional status would allow an early onset of dietary, physical exercise, and, if needed, pharmacological interventions. Exercise prescription for diseases is becoming a standard practice worldwide, and several scientific reports highlight its growing role [35]. The introduction of a program of prehabilitation, including both nutritional and exercise interventions, seems to be justified. Thereby, there are studies that showed that exercise prehabilitation reduced postoperative complications in high-risk patients scheduled to undergo elective colon resection [36]. Likewise, a randomized controlled trial found that multimodal prehabilitation improves functional capacity before and after surgery, enhancing postoperative clinical outcomes [37]. Finally, a systematic review and meta-analysis showed that exercise, nutritional, and multimodal prehabilitation may reduce morbidity after abdominal surgery in patients with cancer [38].

Our study has several strengths: it was a prospective study with a considerable number of subjects and with long-term monitoring, using tools that can be easily used at the hospital and outpatient levels.

All the same, there are potential limitations in our study. It was a single-center, observational study; therefore, the results should be interpreted with caution, and causal links cannot be established. On the other hand, the SPPB can have a ceiling effect affecting patients with an optimal functional status [20]. In this study, 50% of the subjects had a full score on the SPPB, and the effect of the ceiling effect of the SPPB may be present. Finally, the fact of having a systematic nutritional intervention protocol may have influenced the results with a lack of association between malnutrition and clinical outcomes, compared to other studies.

## 5. Conclusions

Low physical performance, assessed by the SPPB and HGS, was elevated in colorectal cancer patients prior to surgery and was associated with postoperative complications and mortality. Assessing patients undergoing colorectal cancer surgery using the SPPB and HGS could help stratify patients at risk of postoperative complications and mortality. Therefore, consideration of preoperative countermeasures against impaired physical performance is necessary.

## Figures and Tables

**Figure 1 nutrients-14-01484-f001:**
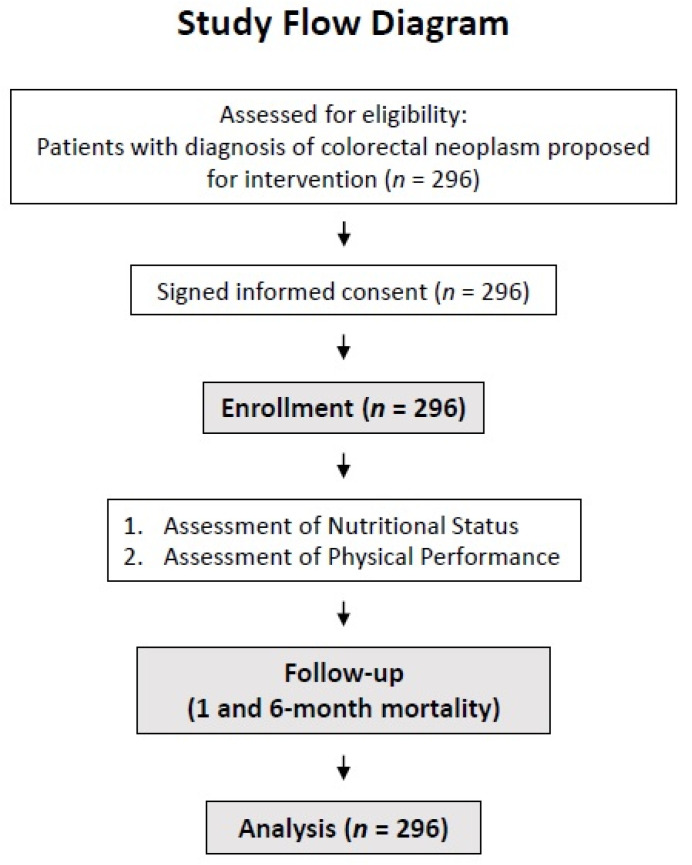
Study flow diagram.

**Table 1 nutrients-14-01484-t001:** General features.

		*n* = 296
Age (years)	mean ± SD (min–max)	68.4 ± 10.2 (30–89)
Sex	*n* (%)	
Men		175 (59.1)
Women		121 (40.9)
Type of cancer	*n* (%)	
Colon		180 (60.8)
Rectum		116 (39.2)
Stage	*n* (%)	
I		39 (13.1)
II		99 (33.5)
III		124 (41.9)
IV		34 (11.5)
BMI (kg/m^2^)	mean ± SD (min–max)	
Men		27.6 ± 5.1 (17.2–47.6)
Women		26.5 ± 5.3 (15.8–46.1)
Surgical complications	*n* (%)	131 (44.2)
Postoperative collection		24 (18.3)
Paralytic ileus		23 (17.6)
Surgical wound infection		23 (17.6)
Suture dehiscence		16 (12.2)
Febrile syndrome		13 (9.9)
Bleeding		12 (9.2)
Other		20 (15.3)
In-hospital exitus	*n* (%)	8 (2.7)
1-month exitus	*n* (%)	9 (3)
6-month exitus	*n* (%)	14 (4.7)

Abbreviations: BMI = body mass index; SD = standard deviation.

**Table 2 nutrients-14-01484-t002:** Short Physical Performance Battery (SPPB) test and hand grip strength.

		*n* = 296
SPPB		
Balance (points)	mean ± SD	3.81 ± 0.48
4 m gait speed (points)	mean ± SD	3.60 ± 0.83
Sit to stand (points)	mean ± SD	3.14 ± 1.14
Total (points)	mean ± SD	10.57 ± 2.07
SPPB < 10 (low physical performance)	*n* (%)	69 (23.3%)
SPPB ≥ 10	*n* (%)	227 (76.7%)
Hand grip strength		
Men	mean ± SD (min–max)	34.01 ± 8.57 (13.3–57.8)
Women	mean ± SD (min–max)	21.03 ± 5.09 (10.6–34)
Low hand grip strength	*n* (%)	58 (19.6%)
Normal hand grip strength	*n* (%)	238 (80.4%)

Abbreviations: SD = standard deviation.

**Table 3 nutrients-14-01484-t003:** Differences between the SPPB and hand grip strength groups.

	SPPB ≥ 10 (*n* = 227) Mean ± SD	SPPB < 10 (*n* = 69) Mean ± SD	*p* Value	Normal Hand Grip Strength (*n* = 238) Mean ± SD	Low Hand Grip Strength (*n* = 58) Mean ± SD	*p* Value
Age (years)	67 ± 9.9	73.2 ± 9.9	<0.001	67 ± 10	74.3 ± 9.2	<0.001
BMI (kg/m^2^)	26.9 ± 4.9	27.7 ± 6.3	0.29	27.5 ± 5.3	25.8 ± 5.1	0.033
Hand grip strength (kg)						
Men	35.2 ± 7.7	25.4 ± 6.5	<0.001	37.3 ± 6.1	21.8 ± 4	<0.001
Women	22.7 ± 4.5	17.1 ± 4.2	<0.001	22.6 ± 4	13.3 ± 1.6	<0.001
Brachial circumference (cm)	28.8 ± 3.8	28.4 ± 4.8	0.49	29.1 ± 3.8	27 ± 4.6	<0.001
Albumin (g/dL)	3.7 ± 0.5	3.5 ± 0.5	0.003	3.7 ± 0.4	3.5 ± 0.6	0.006
CRP (mg/dL)	7.7 ± 11.8	17.9 ± 26	0.003	7.1 ± 9	19.5 ± 29	<0.001
CRP/albumin ratio	2.3 ± 3.9	6.7 ± 11.7	0.001	2 ± 2.7	7.4 ± 12.4	<0.001
Length of stay (days)	11.1 ± 8.9	14.1 ± 9.4	0.018	11.2 ± 9.2	13.8 ± 8.1	0.058

Abbreviations: SPPB = Short Physical Performance Battery; SD = standard deviation; BMI = body mass index; CRP = C-reactive protein.

**Table 4 nutrients-14-01484-t004:** Risk of presenting postoperative complications, adjusted for age, gender, and stage of tumor.

	Crude				Adjusted			
	Odds Ratio	95% CI		*p* Value	Odds Ratio	95% CI		*p* Value
		Lower	Upper			Lower	Upper	
Low SPPB	2.37	1.35	4.17	0.003	2.52	1.35	4.70	0.004
Low hand grip strength	2.77	1.51	5.07	0.001	2.62	1.33	5.13	0.005

Abbreviations: SPPB = Short Physical Performance Battery; CI = confidence interval.

**Table 5 nutrients-14-01484-t005:** Relationship between low physical function and mortality.

	In-Hospital Mortality			1-Month Mortality			6-Month Mortality		
	Normal	Low	*p* Value	Normal	Low	*p* Value	Normal	Low	*p* Value
SPPB	2 (0.9%)	6 (8.7%)	0.021	3 (1.3%)	6 (8.7%)	0.002	2 (2.2%)	9 (13.1%)	<0.001
Hand grip strength	5 (2.1%)	3 (5.3%)	0.187	6 (2.5%)	3 (5.3%)	0.280	8 (3.3%)	6 (10.5%)	0.022

Abbreviations: SPPB = Short Physical Performance Battery.

## Data Availability

Not applicable.

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
