# Peer review of "Poor Physical Performance Is Associated with Postoperative Complications and Mortality in Preoperative Patients with Colorectal Cancer"

_nutrients, 2022, doi:10.3390/nu14071484_

Round 1

Reviewer 1 Report

This is a very interesting article about physical fitness of oncologic patients. I want to congratulate you for your work

Some minor suggestion (pay attention to grammar and synthax rules)

  • line 16 lacks verb
  • line 41: singular not plural
  • line 46 lacks verb
  • line 77 lacks verb
  • Review board? Helsinki declaration?
  • line 84: delete to this effect
  • pay attention to spaces after dot all over the paper
  • Could you improve your figures and tables?
  • I suggest to improve your final paragraph about the importance of physical activity as a drug in patients with different diseases. Please consider the following references (https://www.degruyter.com/document/doi/10.1515/jbcpp-2021-0279/html) 

Author Response

Authors:

Dear Reviewer,

Thank you for giving us the opportunity to improve our article “Poor physical performance is associated with postoperative complications and mortality in preoperative patients with colorectal cancer.”

The various suggestions have been incorporated into the new version wherever applicable. Please find below our responses and the action taken to all the suggestions and comments.

Please see the attachment to check the new version of the manuscript.

Once again, we very much appreciate all the work with the review.

Yours sincerely,

Dr. Francisco José Sánchez Torralvo

Dr. Gabriel Olveira

Reviewer: This is a very interesting article about physical fitness of oncologic patients. I want to congratulate you for your work

Some minor suggestion (pay attention to grammar and synthax rules)

  • line 16 lacks verb
  • line 41: singular not plural
  • line 46 lacks verb
  • line 77 lacks verb
  • Review board? Helsinki declaration?
  • line 84: delete to this effect
  • pay attention to spaces after dot all over the paper

A: Thank you for your appreciation. We have made the suggested changes

  • Could you improve your figures and tables?

A: Thank you for your appreciation. We have improved tables and figures where possible

  • I suggest to improve your final paragraph about the importance of physical activity as a drug in patients with different diseases. Please consider the following references (https://www.degruyter.com/document/doi/10.1515/jbcpp-2021-0279/html) 

A: We very much appreciate your suggestion. We have included a reference on the subject in that paragraph.

Reviewer 2 Report

Dear Authors,

I would like to congratulate you with this study.

On the whole it is interesting study. However, it lacks novelty. Moreover, now the trend is concentrating on prerehabilitation. It is well known that patients with poor nutrition or functional and physical status will perform worse following the surgery. And the  assessment tool does not make the study very novel or adds anything to the scientific body very significant.

Other issues

INTRO

--First reference should be renewed. 

--No further comments - well written

METHODS

--what were the inclusion/exclusion criteria?

--flow chart diagram should be included

RESULTS

--min and max values must be included

--table 4 - you adjusted only for age and sex. But the stage, comorbidities are also very important for the postoperative course. 

DISCUSSION

--well written

--

Author Response

Authors:

Dear Reviewer,

Thank you for giving us the opportunity to improve our article “Poor physical performance is associated with postoperative complications and mortality in preoperative patients with colorectal cancer.”

The various suggestions have been incorporated into the new version wherever applicable. Please find below our responses and the action taken to all the suggestions and comments.

Please see the attachment to check the new version of the manuscript.

Once again, we very much appreciate all the work with the review.

Yours sincerely,

Dr. Francisco José Sánchez Torralvo

Dr. Gabriel Olveira

Dear Authors,

I would like to congratulate you with this study.

On the whole it is interesting study. However, it lacks novelty. Moreover, now the trend is concentrating on prerehabilitation. It is well known that patients with poor nutrition or functional and physical status will perform worse following the surgery. And the  assessment tool does not make the study very novel or adds anything to the scientific body very significant.

A: We very much appreciate your opinion. With the findings of the study, we intend precisely to promote the use of prehabilitation to improve the postoperative aspects of patients with colorectal cancer.

Other issues

INTRO

--First reference should be renewed. 

Thank you for your appreciation. We have renewed the first reference in the new version of the manuscript.

--No further comments - well written

METHODS

--what were the inclusion/exclusion criteria?

All the patients proposed for intervention at the Coloproctology Unit of the Hospital Regional Universitario de Malaga were evaluated. All assessed patients who signed the informed consent were included.  This was the only exclusion criterion and fortunately, all the patients signed it.

--flow chart diagram should be included

Thank you for your appreciation. We have included a flow chart diagram in the new version of the manuscript.

RESULTS

--min and max values must be included

Thank you for your appreciation. We have included min and max values in some descriptive data.

--table 4 - you adjusted only for age and sex. But the stage, comorbidities are also very important for the postoperative course. 

We very much appreciate your suggestion. We have included the adjusted results including the stage of the tumor. We regret not including the adjustment for comorbidities as we do not have enough data in this regard.

DISCUSSION--well written

Reviewer 3 Report

The paper  aproaches  an important aspect  of cancer patients . It colud be  implementeted with the  evaluation  of  the  hydration status  that completes the  nutritional evaluation . Expecially the PA( angle Phase )  is normally realted  to the  increased risk  of mortality . In addition considering the patients are all overweight , a definition  of FF( Fatty Mass) and FFM ( fatty free mas) coulb be important . The  author shoul  clarify , in any  case, some  clinical  aspect of these missin  g data in the paper . 

Author Response

Authors:

Dear Reviewer,

Thank you for giving us the opportunity to improve our article “Poor physical performance is associated with postoperative complications and mortality in preoperative patients with colorectal cancer.”

Once again, we very much appreciate all the work with the review.

Yours sincerely,

Dr. Francisco José Sánchez Torralvo

Dr. Gabriel Olveira

The paper  aproaches  an important aspect  of cancer patients . It colud be  implementeted with the  evaluation  of  the  hydration status  that completes the  nutritional evaluation . Expecially the PA( angle Phase )  is normally realted  to the  increased risk  of mortality . In addition considering the patients are all overweight , a definition  of FF( Fatty Mass) and FFM ( fatty free mas) coulb be important . The  author shoul  clarify , in any  case, some  clinical  aspect of these missin  g data in the paper . 

Authors: We very much appreciate your suggestion. Unfortunately, we do not yet have data regarding body composition. This will be studied in future papers.

Round 2

Reviewer 2 Report

Dear Authors,

Although the article improved a lot, I still do not think that the study design is suitable for this journal.